# The Use of Fluorescence Spectrometry to Determine the Botanical Origin of Filtered Honeys

**DOI:** 10.3390/molecules25061350

**Published:** 2020-03-16

**Authors:** Aleksandra Wilczyńska, Natalia Żak

**Affiliations:** Gdynia Maritime University, Department of Commodity Science and Quality Management, ul. Morska 81-87, 81-225 Gdynia, Poland; n.zak@wpit.umg.edu.pl

**Keywords:** filtered honey, botanical origin, fluorescence spectrometry

## Abstract

The aim of this study was to determine whether fluorescence spectrometry can be used to identify the botanical origin of filtered honeys. Sixty-two honey samples with different botanical origins, both filtered and unfiltered, were investigated in order to examine their fluorescence spectra. The results showed that individual honey varieties have different fluorescence spectra, and the filtration process had no impact on these spectra. The results suggest that fluorescence spectroscopy may be a useful method to identify the botanical origin of filtered honeys.

## 1. Introduction

Honey is a valuable, natural food product produced by *Apis mellifera* from plant juices (honeydew) and flower nectar. Both are processed by bees and enriched with their scent-gland secretions. It is a product of the highest quality, not obtainable by any other means. The quality of honey depends on environmental conditions, the climate, the beekeeper’s interference in the production process, and the method of its collection and storage. However, the chemical composition and sensory characteristics of honey depend mainly on its botanical origin—the type and species of the plant that it comes from. Additionally, the specific chemical composition of different honey types determines their therapeutic properties [1,2,3,4].

Due to its high price, natural honey is often adulterated [5]. The results of the honey quality control indicate that many producers use misleading practices regarding the expected nutritional and health properties of honey. The most common adulteration techniques involve feeding sugar syrup to the bees during the nectar flow, or adding sugar to honey to increase the final product volume. [6]. However, the most common practice, recently, is the falsification of the botanical origin of honey, consisting in declaring honey as unifloral, when it is the cheapest multi-flower honey or honey imported from outside the European Union, e.g., from China.

The honey filtration process also helps to falsify it. According to Council Directive 2001/110/EC relating to honey, filtered honey is: “honey obtained by removing foreign inorganic or organic matter in such a way as to result in the significant removal of pollen”. As a result of honey filtration, glucose crystals, impurities, yeast cells, wax fragments, and dyes are removed. However, the most important fact is that most pollen grains are lost as a result of filtration. Therefore the filtration process changes slightly the organoleptic properties of honey (color, taste, smell), as well as its chemical composition and biological properties [7,8]. This prevents the honey variety from being correctly identified.

The identification of the biological origin of honey is very important, because it determines the properties of individual honeys. The most common reference method for identifying polyfloral honey types, recommended by Codex Alimentarius and other honey standards, is the time-consuming palynological analysis. However, in some cases, the percentage of pollen is not always decisive, because the production of pollen and nectar by flowers is not always simultaneous, varying between countries and even within the same country, according to the geographical area [9]. Moreover, a palynological analysis is protracted and involves dedicated experts. The authenticity and health benefits of different honey types can also be determined based on their sensory parameters and chemical composition: amino acids, sugars, polyphenols, volatile compounds, etc. [10,11,12]. The combination of the multi-component analysis and chemometric techniques is also increasingly used [13,14,15,16,17,18,19,20]. However, due to the fact that the filtration process affects the chemical composition of honey, it is almost impossible to use all of these methods to properly identify the botanical origin of filtered honeys.

The authenticity of different honey types may also be ascertained by infrared spectroscopy and fluorescence spectroscopy [21,22]. The advantage of fluorescence spectroscopy is the high sensitivity and specificity of the classification. Fluorescence spectroscopy requires only a minimal sample preparation. The results of the above-mentioned studies confirmed that single synchronous fluorescence spectra of different honeys differ significantly because of their distinct physical and chemical characteristics, and provide sufficient data for the clear differentiation among honey groups. The studies demonstrated that this method is a valuable and promising technique for honey authentication. According to Ruoff [23], honeys are well known to contain numerous fluorophores such as polyphenols and amino acids. Some of them have been proposed as tracers for unifloral honeys—ellagic acid for heather honey, hesperetin for citrus honey, phenylalanine and tyrosine, which have been found to be characteristic for lavender honey and allowed a differentiation from eucalyptus honey, and tryptophan and glutamic acid, which have been shown to be useful for the differentiation between honeydew and blossom honeys. Due to the presence of such strong fluorophores, fluorescence spectroscopy may be helpful for authenticating the botanical origin of honey. Therefore, the aim of our study was to determine whether fluorescence spectra can also be used to identify the botanical origin of filtered honeys.

## 2. Results and Discussion

The emission spectra (excitation: 200–450 nm; emission: 260–560 nm) considered in this investigation allowed the study of the fluorescence of honey samples and the variation of their botanical origins and filtration. The EEM spectra were measured for the tested honeys. Figure 1 shows the results as contour maps, after removing Rayleigh scattering for 62 samples of different honey types: multifloral, honeydew, acacia, goldenrods, rape, phacelia, lime, buckwheat, and dandelion.

In Figure 1, it appears that each honey type exhibited a specific fluorescence spectrum. It was concluded that emission spectra (260–560 nm) are fingerprints allowing a good identification of the botanical origin of honeys. It can be seen that the results were consistent with the results of other researchers [22,24,25,26,27,28,29].

It could be seen that dandelion and goldenrods fluoresced very strongly over the emission range of 380–540 nm and the excitation range of 300–435 nm. On the other hand, rape honey samples had an almost negligible emission in this spectral region. But the intensity of the fluorescence was very strong over the emission range of 320–410 nm and the excitation range of 250–310 nm. It could also be seen that acacia, buckwheat, and honeydew honeys were characterized by two very intense emission sources (acacia: 320–385 nm and 390–500 nm, buckwheat: 310–360 nm and 380–450 nm, and honeydew: 320–385 nm and 380–525 nm).

Figure 2 shows the synchronous cross-sections of these spectra obtained at Δλ = 100 nm for honey samples and the variation of their botanical origins. Similar results were presented by Gębala [30] and Lenhardt et al. [22].

Figure 3 shows the fluorescence spectra of the same honey samples, but subjected to a filtration process. Statistical analysis confirmed that the filtration process does not affect the shape of the spectra (*F* = 3.65, *p* = 0.056), while the differences in the spectra of honeys of different botanical origins were statistically significant (*F* = 8.59, *p* = 0.00).

The spectra of all tested honeys in connection with the filtration process are characterized by the presence of the same characteristic emission strip of variable intensity, as it can be seen in Figure 4a–i. In Table 1, the mean emission intensities over characteristic spectral regions for filtered and unfiltered honeys of different types are summarized.

Filtered honeys exhibited higher fluorescence intensities for acacia honey. It was possible to observe characteristic changes in the synchronic spectrum of the filtered acacia honey in relation to the spectrum of the unfiltered acacia honey (Δλ = 100 nm): increased intensities in the short-wave strip, increased intensity in the strip of intermediate range, and no change in the long-wave spectrum. There was a noticeable tendency for a rise to three emission excitances: 230 nm for unfiltered and filtered honeys, 275 nm and 340 nm for the filtered honey, and 280 nm and 335 nm for the unfiltered honey. Considering the filtration, an increase in the intermediate strip emission intensity (from 235 to 430 nm) was observed (Figure 4a and Table 1).

In the case of the synchronic spectrum of the phacelia honey (Δλ = 100 nm) there were three emission excitations: 230 nm, 275 nm, and 355 nm. Considering the filtration, an increase in the intensity of the emission line was observed from 250 to 410 nm (Figure 4b and Table 1).

Figure 4c shows the set of synchronic spectra for buckwheat honeys obtained at Δλ = 100 nm. It can be observed that the spectra of all studied buckwheat honeys were characterized by the presence of the same emission strips. Their intensity is different only in the range of 325 to 415 nm. In this range, the intensity of unfiltered honeys increased significantly (Figure 4c and Table 1).

The same results were obtained for the set of synchronic spectra for the lime honeys obtained at Δλ = 100 nm. It could be observed that the spectra of all studied lime honeys were characterized by the presence of the same emission strips and intensity. The only exception was the intensity of the emission in the range of 275 to 365 nm. In this range, the intensity of unfiltered honeys increased slowly (Figure 4d and Table 1).

Filtered honeys included lower fluorescence intensities for multifloral honeys. It was possible to observe characteristic changes in the synchronic spectrum of the filtered multifloral honey in relation to the spectrum of the unfiltered acacia honey (Δλ = 100 nm): increased intensities in the short-wave strip, increased intensity in the strip in the intermediate range, and no change in the long-wave spectrum. There was a noticeable tendency for up to three emission excitances: 230 nm, 275 nm, and 360 nm. The fluorescence intensity of filtered and unfiltered honeys was the same from 420 to 450 nm (Figure 4e and Table 1).

Filtered honeys included higher fluorescence intensities for rape honeys. It was possible to observe characteristic changes in the synchronic spectrum of filtered rape honeys in relation to the spectrum of unfiltered acacia honey (Δλ = 100 nm): no change in the short-wave strip (225 nm), increased intensity in the strip in the intermediate range, and in the long-wave spectrum. There was a noticeable tendency for up to two emission excitances: 230 nm, and 275 nm. The third maximum (335 nm) was observed to be flatter (Figure 4f and Table 1).

Figure 4g and Table 1 show the set of synchronic spectra for dandelion filtered and unfiltered honeys obtained at Δλ = 100 nm. It can be observed that the spectra of all studied dandelion honeys were characterized by the presence of the same emission strips. Their intensity was different only in the range of 200 to 285 nm. The third maximum (350 nm for unfiltered honeys and 355 nm for filtered honeys) was observed. In this range, the intensity of unfiltered honeys gently increased.

It was possible to observe characteristic changes in the synchronic spectrum of the filtered honeydew honey in relation to the spectrum of the unfiltered acacia honey (Δλ = 100 nm). The short-wave and intermediate range strip spectrum (from 200 to 380 nm) of the filtered honeys included higher fluorescence intensities than unfiltered honeys. There was a noticeable tendency for up to three emission excitances: 230 nm, 280 nm, and 360 nm. In addition, no changes could be observed in the long-wave spectrum including filtered and unfiltered honeys, from 380 to 450 nm (Figure 4h and Table 1).

Figure 4i and Table 1 show the set of synchronic spectra for goldenrod honeys obtained at Δλ = 100 nm. It can be observed that the spectra of all studied goldenrods honeys were characterized by the presence of the same emission strips. Their intensity was different only in the range of 320 to 415 nm. In this range, the intensity of unfiltered honeys increased significantly. The fluorescence intensity of filtered and unfiltered honeys was the same from 420 to 450 nm—no changes in the long-wave spectrum. 

An appropriate classification and confirmation of the honey’s authenticity is extremely important, because the health-promoting effect of honey is related to its chemical composition, and hence, to its botanical origin. The obtained results have once again confirmed that fluorescence spectrometry is an excellent method for the fast and nondestructive identification of the botanical origin of honey, and these results were consistent with those of other researchers [26,27,28,29,30,31]. However, it has been demonstrated for the first time that fluorescence spectrometry can be used also to determine the botanical origin of filtered honeys. As mentioned in the introduction, the main fluorophores in honey are phenolic compounds and aromatic amino acids. The EEMs of honey are characterized by a spectral region characterized by high emission intensities, and these emissions come from these compounds. The results obtained in this study confirm our previous observations that the filtration process does not change the phenolic content of honey [8]. Therefore, the spectra of filtered honeys differed slightly in the fluorescence intensity, while the shape of the spectra did not change.

## 3. Materials and Methods

### 3.1. Samples

Sixty-two honey samples of 9 different botanical origins (multifloral: 12, honeydew: 12, acacia: 3, goldenrods: 3, rape: 9, phacelia: 5, lime: 5, buckwheat: 8, dandelion: 5) were analyzed to evaluate the effect of filtration on the fluorescence spectra. The honeys, provided by a local beekeepers’ association from the Pomeranian province, were harvested in 2017. Each sample was available as unfiltered and originally filtered forms, to compare the honeys directly.

The samples (100–150 g) were subjected to a filtration process—for this purpose they were rapidly heated to 45 °C for 5–10 min, to reduce the honey viscosity and dissolve any crystals. The honey was mixed thoroughly, then filtered through Schott filters (Duran, Mainz, Germany) with a pore size <100 μm, under a pressure of 0.3–0.4 MPa. The honey was then cooled down. The filtered and unfiltered samples were stored without light at room temperature until the analysis (no longer than 48 h).

### 3.2. Methods

Fluorescence spectra were determined by a method patented by Gębala and Przybyłowski [30,31]. The studies were carried out using a set-up based on the Fluorescence Spectrophotometer F-7000 Hitachi, Japan. A special adapter was built for it, in order to change its traditional measurement range (Figure 5.). During the measurement, the fluorescence intensity was measured from the surface of the sample, where the excitation radiation falls on. A reflective geometry enable to eliminate the effects of the internal filter associated with high absorbance of the sample—weakening of the fluorescence intensity due to the absorbance of excitation and the radial radiation emitted [30,31].

The dimensional fluorescence spectra were measured at room temperature and daylight. Honey samples were liquefied at 40 °C and pipetted into 0.5 mL quartz cuvettes before measurements. The fluorescence spectra were obtained by recording the emission spectra (from 220 to 560 nm with a 10-nm step) corresponding to excitation wavelengths ranging between 200 and 450 nm (with a 5-nm step), and automatically normalized to the excitation intensity by the instrument. The sensitivity of the excitation and emission measurements was stated at a voltage equal 600 V. The difference between the fluorescent light wavelength (γ_F_) and the excitation light wavelength (γ_w_) was preferably 100 nm [30,31].

The fluorescence spectra were normalized by reducing the area under each spectrum to a value of 1 [24]. This was to reduce scattering effects and compare the investigated honey samples.

All analyses were done in triplicate. The final results are presented as a set of numerical data in the form of contour maps (excitation emission (EEM)) and the synchronous cross-sections of these spectra were obtained at Δλ = 100 nm for honey samples.

To determine the effect of the filtration or botanical origin on the shape of the spectra, one- and two-way analysis of variance (ANOVA) were used. Statistical hypotheses were verified at the significance level of *p* = 0.05.

## 4. Conclusions

1. The methodology proposed here allowed honey samples to be distinguished based on their different botanical origins, by using the simple and fast analysis of their fluorescence spectra.

2. The fluorescence spectra were the same for the filtered and unfiltered honeys, but the intensity of the fluorescence was different. This means that fluorescence spectra can also be used to identify filtered honeys.

## Figures and Tables

**Figure 1 molecules-25-01350-f001:**
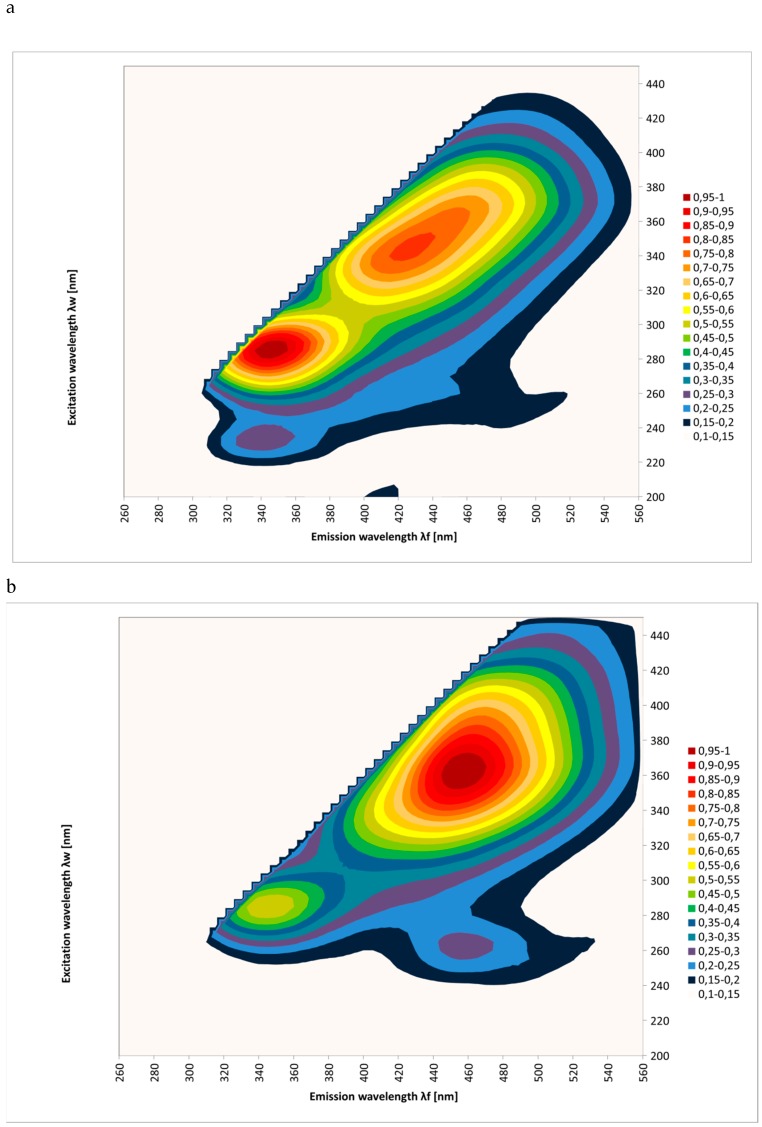
Excitation emission (EEM) spectra of different botanical origins of honey (**a**–acacia, **b**–phacelia, **c**–buckwheat, **d**–lime, **e**–multifloral, **f**–rape, **g**–dandelion, **h**–honeydew, and **i**–goldenrods). Source: own research.

**Figure 2 molecules-25-01350-f002:**
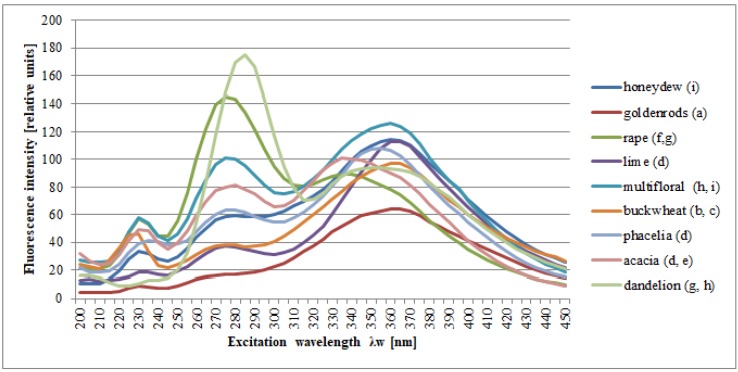
Normalized fluorescence emission spectra of different botanical origins of unfiltered honeys; the letters (**a**–**i**) indicate homogeneous groups, separated as a result of post-hoc analysis (Duncan’s test). Source: own research.

**Figure 3 molecules-25-01350-f003:**
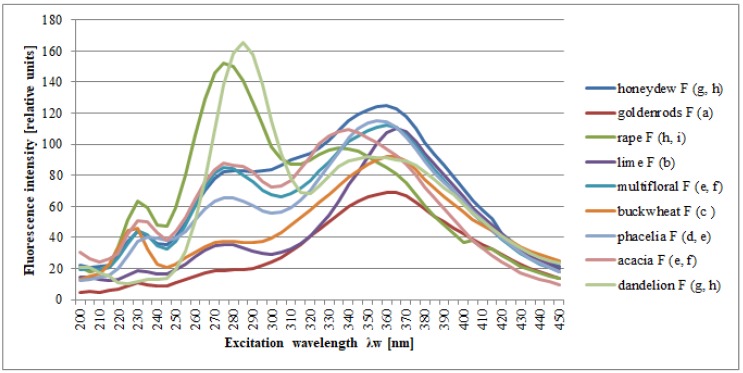
Normalized fluorescence emission spectra of different botanical origins of filtered honeys; the letters (**a**–**i**) indicate homogeneous groups, separated as a result of post-hoc analysis (Duncan’s test). Source: own research.

**Figure 4 molecules-25-01350-f004:**
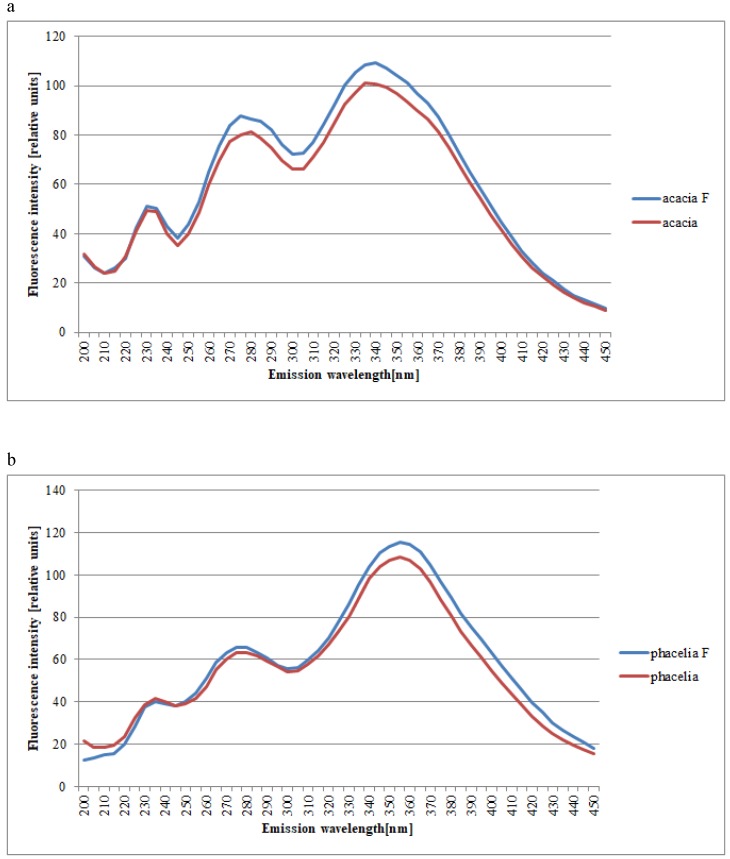
Normalized fluorescence emission spectra of different botanical origins of filtered and unfiltered honeys (**a**–acacia, **b**–phacelia, **c**–buckwheat, **d**–lime, **e**–multifloral, **f**–rape, **g**–dandelion, **h**–honeydew, and **i**–goldenrods) F-filtered and unfiltered. Source: own research.

**Figure 5 molecules-25-01350-f005:**
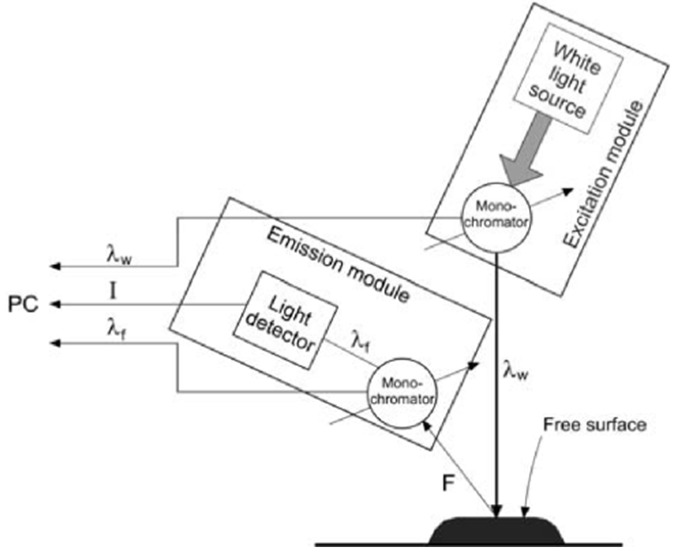
Scheme of surface fluorescence measurement. Source: [30,31].

**Table 1 molecules-25-01350-t001:** Mean emission intensities over characteristic spectral regions for filtered and unfiltered honeys of different types.

Spectral Region	Excitation	Type of Honey	Kind of Processing	Mean Intensities of Fluorescence	SD
1st	200	dandelion	unfiltered	21.19	1.69
1st	230	honeydew		33.34	3.43
1st	230	goldenrods		8.96	2.62
1st	230	rape		45.78	2.82
1st	230	multifloral		58.15	3.98
1st	230	acacia		51.04	3.50
1st	235	phacelia		41.46	3.62
2nd	275	rape		37.59	2.26
2nd	275	lime		77.52	9.15
2nd	275	multifloral		100.96	6.17
2nd	275	buckwheat		37.59	2.21
2nd	275	phacelia		63.43	4.26
2nd	280	honeydew		59.42	6.32
2nd	280	acacia		86.65	5.83
2nd	285	dandelion		110.27	6.28
3rd	335	rape		67.70	2.46
3rd	335	acacia		108.37	4.63
3rd	355	dandelion		106.82	2.46
3rd	355	phacelia		108.45	3.96
3rd	360	honeydew		114.42	4.69
3rd	360	multifloral		126.41	4.19
3rd	360	buckwheat		91.98	2.90
3rd	365	goldenrods		64.49	4.13
3rd	365	lime		107.44	4.13
1st	200	dandelion	filtered	25.41	1.69
1st	230	honeydew		43.98	2.96
1st	230	goldenrods		10.66	4.94
1st	230	rape		45.59	3.06
1st	230	multifloral		44.56	3.79
1st	230	acacia		37.89	3.78
1st	235	phacelia		40.16	3.05
2nd	275	rape		38.65	2.21
2nd	275	lime		35.35	1.74
2nd	275	multifloral		84.70	6.76
2nd	275	buckwheat		37.91	2.20
2nd	275	phacelia		65.57	3.96
2nd	275	acacia		80.11	5.39
2nd	280	honeydew		82.98	5.58
2nd	285	dandelion		105.81	6.07
3rd	335	rape		72.68	2.78
3rd	340	acacia		100.66	3.69
3rd	350	dandelion		103.96	2.71
3rd	355	phacelia		115.21	3.68
3rd	360	honeydew		124.82	4.69
3rd	360	goldenrods		69.06	6.26
3rd	360	multifloral		111.97	3.89
3rd	360	buckwheat		97.24	3.58
3rd	365	lime		110.20	4.75

SD–standard deviation. Source: own research.

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
