# Peer review of "The Use of Fluorescence Spectrometry to Determine the Botanical Origin of Filtered Honeys"

_molecules, 2020, doi:10.3390/molecules25061350_

Round 1

Reviewer 1 Report

The work is interesting and the topic deserves to be investigated, being the fluorescence technique largely applied in the food quality and traceability field. 

I have some remarques to the authors:

1) the synchrounous florescence technique should be a bit defined for the non-expert reader

2) the sensitivity of the excitation and emission measurements should be stated

3) The authors fixed the wavelenght difference for the SF at 100 nm, with no explication. It is well known that the SF results largely depend on the Delta Lambda, and that different Deltas allows individuating different emitting species, in case they are present.

4) The choice of the colours for the EEM must be revised. i understand that it recalls the honey colour but it is not useful to the presentation of the data

5) In general, the quality of the spectra should be improved. 

6) The x-axis of the SF spectra is excitation, not emission

7) The authors just describe the differences, but did no try to assign them to any specific chemical composition of the samples

8) The authors stated that they analyzed different samples for each variety. I'd like to see if the proposed method is able to differentiate among the varieties. Maybe the analysis by means of principal component or the like could be exploited to this purpose (see for example the wine case: Spectrochim. Acta A Mol. Biomol. Spectrosc., 148 (2015), pp. 131-137, or Spectrochim. Acta A Mol. Biomol. Spectrosc., 241 (2019), pp 378-383)

Author Response

At the beginning I would like to thank you for your thorough review and respond to your comments:

1) The synchronous fluorescence technique has been refined and defined in a more understandable and comprehensive way.

2) The sensitivity of the excitation and emission measurements was stated at voltage equal 600V.

3) Research was carried out using the method patented by Gębala and Przybyłowski who stated, that the difference between the wavelength of fluorescent light (γF) and the wavelength of excitation light (γw) is most favorable at 100 nm.

4) The choice of colours for EEM has been changed to a more readable one for presentation.

5) The quality of the spectra has also been improved.

6) The x-axis descriptions of SF spectra have been corrected, where the X-axis of SF spectra is excitation, not emission.

7) The authors referred to this in the discussion of the results.

8) The subject of our analysis were the average spectra of honeys belonging to different varieties, we did not analyse the spectra of individual honeys. We have shown that each honey variety has a different spectrum shape. This was confirmed by the results of ANOVA analysis of variance. Since the purpose of our work was to show that filtration does not affect the shape of the spectra, we did not conduct further analysis to prove that proposed method is able to differentiate among the varieties.

Reviewer 2 Report

This work is on the spectroscopic characterization of honey samples with different botanical origin, motivated by the need to have a screening technique to identify adulterated or fake commercial product. The approach followed by authors is simple as it consists in obtaining EEM spectra. Based in the results, it is claimed that fluorescence spectrum can be used to identify the botanical origin of honey. This work is of interest and can be considered for publication after major revision.

1) Introduction. Literature is vast in the number of studies on honey authentication or screening the botanical origin. This section must be edited to state more clearly previous achievements and limitations of fluorescence-based spectroscopies. In view of this revision of literature at the introduction, the reader would rapidly identify the contribution of the present work.

2) Materials and methods. When EEm spectra are obtained, special care must be paid to absorption effects. It seems that samples were measured without dilution, which could create artifacts on the measurements. Please provide absorption spectra of the measured samples and discuss possible inner effects.  This is an important issue that authors have not to underestimate. A better description of samples and experimental set-up would help.

3) Readers would appreciate if a brief description on the method described in Ref [19-20] is presented. That method is used in this work to determine spectra.

4) Figure 3 is not presented neither Table 3 mentioned in figure caption of Fig 2.

5) The sample. It consists of 9 honeys of different botanical origin. The number of samples per type of honey is not the same. As the samples are from the same beekeeper, it is not clear if testing various samples of a honey of the same botanical origin, but different beekeeper would lead to the same results. This would imply another work, I agree, but please discuss briefly the implications of screening samples from different beekeeper.  Reader assume that samples are from different beehives. Please clarify.

6) A table with the statistics of the samples should be provided. Even though ANOVA analysis was performed to support the claim that the fluorescence spectra are statistically different, it is not clear to the reader the range of uncertainties in these measurements. Please for each type of honey provide information with standard deviation error.

7) Line 111. Is it possible to obtain p=0? 

8) Pos-hoc analysis was performed but I could not find a summary of the results in a table.  Reader would appreciate if tables are added with this information.

9) The differences in the spectra are already observed in figure 1. Authors describe globally these differences in the main text. As the authors claim that each botanic origin would have associated a fluorescence fingerprint, figures of merit should be presented.  PCA or other technique can be used to reduce the number of variables presented in Figure 1.

10) Introduction. It is mentioned that multicomponent analysis cannot be used to identify the botanical origin of honey. It seems that if the data presented in this work is processed with multicomponent analysis honeys can be identified. Please discuss. If multicomponent analysis works in this case, authors could present some results as an alternative to analyze data.

11) Please Edit figure 1 to have bigger and readable fonts in x and y axis.

12) Discussion should include also some comments on the results and treatment of data on the context of previous reports. Notice that one disadvantages of EEM is that two monochromators are required which make the technique not easy for an inexpensive and simple screening of samples.

Author Response

At the beginning I would like to thank you for your thorough review and respond to your comments.

1) We followed your recommendations - in the introduction we have included several recent publications on the identification of botanical origin of honey.

2) We use the reflective optical system. It is used for measurements of solid samples and  with high absorbance. During the measurement, the fluorescence intensity is measured from the surface of the sample - where the excitation radiation falls on. Reflective geometry enable to eliminate the effects of the internal filter associated with high absorbance of the sample - weakening of the fluorescence intensity due to the absorbance of excitation and radial radiation emitted.

3) The synchronous fluorescence technique has been refined and defined in a more understandable and comprehensive way.

4) I apologize for the lack of Figure 3. It was lost during the editing of the text. Figure 3 was added to the work.

5) Honey samples came from various apiary located from the Pomeranian province. This has been corrected in the text.

6) The results of the statistical analysis were given in the text as F and P values. Additionally we added table 1 in which the mean emission intensities over characteristic spectral regions for filtered and unfiltered honeys of different types are summarized.

7) Yes, it is possible due to the rounding of the very small p-value. In fact it was 0.00005, but we shortened it to 0.0

8) The results of the post-hoc analysis were presented in Figures 3 and 4 as letter designations of homogeneous groups.

9) The purpose of our work was to show that filtration does not affect the shape of the spectra, we did not conduct further analysis to prove that proposed method is able to differentiate among the varieties. We tried to make a PCA analysis, but it did not give unequivocal, easy to interpret results, therefore we resigned from showing them.

10) We did not say, that multicomponent analysis cannot be used to identify the botanical origin of honey. We said, that the identification of botanical origin of filtered honeys is almost impossible.

11) The quality of the spectra has also been improved. The choice of colours for EEM has been changed to a more readable one for presentation.

12) We have included additional explanations in the chapter Results and discussion.

Reviewer 3 Report

the method maybe patented, still give a short explanation of its foundations, pls.

in my version there was no figure 3, needs to be added

the comparison of the spectra depends on the normalization procedure which was not detailed in the methods, needs to be explained

the conclusion of "botanical difference" is well supported by figs 1 and 2

the conclusion of "no filter effect" is somewhat strange after two pages of discussions of differences in fig 4. pls, show the statistical analysis that allows to make this statement.

Author Response

At the beginning I would like to thank you for your thorough review and respond to your comments.

1) The synchronous fluorescence technique has been refined and defined in a more understandable and comprehensive way.

2) I apologize for the lack of Figure 3. It was lost during the editing of the text. Figure 3 was added to the work.

3) The description of the method has been extended.

4) We have included additional explanations in the chapter Results and discussion. We have shown that each honey variety has a different spectrum shape and that the spectra of filtered and unfiltered honeys of the same varieties do not differ significantly. This was confirmed by the results of ANOVA analysis of variance, the values of F and p was given in the text.

Round 2

Reviewer 1 Report

The authors fulfil the previous requirements, although their choice for color map of EEM plots is, in my opinion, worse than the original one. I'd like to suggest to use typical color map, such as rainbow map from blue to red.

As for the overall manuscript I think that it deserves to be published.

Author Response

Dear Reviewer,

thank you once more for taking time to review our article. We we tried to change the colors of the spectra to make them more readable,. We hope you will be satisfied.

Reviewer 2 Report

In my opinion this revised version of the manuscript can be accepted for publication

Author Response

Thank you for revision of our article.

Reviewer 3 Report

1.you state now that "Honey
 samples were liquefied at 40°C and pipetted into 0,5 ml quartz cuvettes"

does the honey stay liquid till after measurement?

how do you wash the cuvettes? (important regarding cross-contamination)

2. in the introduction, you state that samples are different "varying between
countries and even within the same country according to the geographical area". could you please show the extend of this variability by a figure with different samples of nominatively the same plant origin ?

3.you introduced "homogeneous groups, separated as a result of post-hoc analysis (Duncan's test)" in the legend of Figs 3 and 4.

what are these? how are they  obtained?  can you show their spectral characteristics?

4. you use the word "long-term" (time) where "long-wave" (wavelength) should be used

5. in line 117, fig1 should read fig 2

Author Response

Dear Reviewer,

thank you very much for the assessment. We would like to refer to your questions:
1. the analytical procedure lasts about 8 min, also the honeys were not able to crystallize again. We wash  the cuvettes each time with warm water, then rinsed with distilled water and then with methanol - to speed up drying, We did not use any cloths or paper towels.

2. We quoted this statement from: Juan-Borrás, M.; Domenech, E.; Hellebrandova, M.; Escriche, I. Effect of country origin on physicochemical, sugar and volatile composition of acacia, sunflower and tilia honeys. Food Res. Int., 2014, 60, 86–94. We have not studied it ourselves, and therefore we cannot show it in the figures.

3. Total and synchronic spectra can be presented in the form of a series of many numbers and it was in this form that they were subjected to analysis of variance and then post-hoc tests. We use the Statistica software. This is a standard procedure to data analysis.

Including spectra of honeys of different varieties in one homogeneous group (marked with one letter) means that they do not differ significantly from each other. However it is very difficult to show this in the spectra - it would be necessary to superimpose the spectra on each other, which would make them completely illegible. Letters are usually used either in tables or in figures to show homogeneous groups.

4. Thank you for this comment, we followed it.

5. Thank you for this comment, we followed it.